# Effects of High Magnetic Fields on the Diffusion of Biologically Active Molecules

**DOI:** 10.3390/cells11010081

**Published:** 2021-12-28

**Authors:** Vitalii Zablotskii, Tatyana Polyakova, Alexandr Dejneka

**Affiliations:** 1Department of Optical and Biophysical Systems, Institute of Physics of the Czech Academy of Sciences, 18221 Prague, Czech Republic; polyakova@fzu.cz (T.P.); dejneka@fzu.cz (A.D.); 2International Magnetobiology Frontier Research Center, Hefei 230031, China

**Keywords:** magnetic field, molecular diffusion, drug diffusion, hemoglobin, red blood cells

## Abstract

The diffusion of biologically active molecules is a ubiquitous process, controlling many mechanisms and the characteristic time scales for pivotal processes in living cells. Here, we show how a high static magnetic field (MF) affects the diffusion of paramagnetic and diamagnetic species including oxygen, hemoglobin, and drugs. We derive and solve the equation describing diffusion of such biologically active molecules in the presence of an MF as well as reveal the underlying mechanism of the MF’s effect on diffusion. We found that a high MF accelerates diffusion of diamagnetic species while slowing the diffusion of paramagnetic molecules in cell cytoplasm. When applied to oxygen and hemoglobin diffusion in red blood cells, our results suggest that an MF may significantly alter the gas exchange in an erythrocyte and cause swelling. Our prediction that the diffusion rate and characteristic time can be controlled by an MF opens new avenues for experimental studies foreseeing numerous biomedical applications.

## 1. Introduction

Currently, numerous experimental techniques allow the generation of magnetic fields in the range of thousands of teslas. The world’s strongest resistive 41.4–45 tesla magnets, capable of running continuously, were engineered at the National High Magnetic Field Laboratory at Florida State University (2017) and at the High Magnetic Field Laboratory Hefei Institutes of Physical Science, Hefei, China. Static magnetic fields with a magnetic induction of 17.6–21 T are used in MRI instruments for better image resolution and more accurate diagnoses [1,2]. Using high magnetic fields provides new opportunities and excellent experimental conditions for physics, magnetochemistry, material sciences, and life sciences. Research into the biological effects of high magnetic fields is interesting and inspiring, but it is still in its initial stage.

Diffusion is one of the most pervasive processes that controls many mechanisms in cell machinery. The diffusion rate governs the characteristic time scale for intracellular processes, which play pivotal roles in many cell functions such as regulation of cell membrane potential, cell motility, division, gas exchange, intracellular transport, and cell signaling. Thus, since diffusion is often the dynamic basis for a broad spectrum of different intracellular processes, cell functions could be modified by a proper tuning of the diffusion rate of some diffusing species, e.g., tuning of the diffusion coefficient with a magnetic field.

The effects of a magnetic field on diffusion has been a topic of investigation since early observations of the effects of static uniform and non-uniform magnetic fields of diffusion on paramagnetic species [3,4]. The theoretical results [4] predict that, in solution, the Lorentz force can affect the diffusion of univalent ions, but the threshold magnetic field is extremely high, approximately 5.7 × 10^6^ T.

Interactions between a magnetic field and living cells may result in the appearance of a variety of biomagnetic effects at the cellular and organism levels [5,6,7,8,9,10,11,12]. Many intriguing mechanisms have been suggested to explain biomagnetic effects [13,14,15,16,17,18]. The biological effects of high and ultrahigh magnetic fields are of particular interest; for example, a 9.4 T static MF suppress lung cancer growth [12]; a 10.5 T MRI static MF affects human cognitive, vestibular, and physiological functions [19]; a 16.4 T static MF results in long-term impairment of the vestibular system in mice [20]; a 27 T static MF changes the orientation and morphology of mitotic spindles in human cells [21]. The level of oxytocin in mice brain is upregulated by an 11–33 T static MF [22]. An important but poorly understood factor that may affect intracellular processes and chemical reactions is the role of magnetic fields in the diffusion of biologically active molecules. In this work, we propose the mechanisms of the magnetic control of diffusion and discuss biological consequences related to the diffusion of some biologically active molecules in static uniform magnetic fields.

Let us start with a brief description of the relevant magnetic forces that may affect diffusion in cells. When a static magnetic field is applied to cell systems, three types of magnetic forces can act on subcellular components, molecules, and ions: (i) the Lorentz force, ***F_L_*** = *q[**vB**]* (where ***B*** is the magnetic induction, *q* is the ion electric charge, and ***v*** is its velocity); (ii) the magnetic gradient force, *F*_∇*B*_∝∇*B*^2^ [23,24,25] (when the magnetic field is uniform (∇*B* = 0), the magnetic gradient force is zero); (iii) the concentration-gradient magnetic force, *F*_∇*n*_∝*B*^2^∇*n* [26,27,28,29,30] (where ∇*n* is the gradient of the concentration of diamagnetic and paramagnetic species, and ∇ is the differential operator nabla).

The concentration-gradient magnetic force will be considered as the driving force in our model, describing the MF’s effect on diffusion in cells. In fact, living cells are far from having a thermodynamic equilibrium, and hundreds of diamagnetic and paramagnetic species inside cells may have very large concentration gradients [31]. Thus, in the presence of a high MF, a relatively large concentration-gradient magnetic force can be operative in cells. In cells, this force can redistribute paramagnetic free radicals, such as O_3_, NO, and NO_2_, and the molecules FeCl_3_ and O_2_, thereby altering the cell machinery and changing cell fate.

Here, we demonstrate how the concentration-gradient magnetic force can drive diffusion of paramagnetic and diamagnetic species in cells. We show that even though the concentration-gradient magnetic forces exerted on diffusing molecules are small, they may play an important role in the presence of sufficiently high magnetic fields.

The remainder of the paper is organized as follows. In Section 2.1, we provide the basic formulas for the concentration-gradient magnetic force. In Section 2.2, using calculations of the species concentration as a function of both coordinate and time, we show that a high MF can change the characteristic time of diffusion, accelerating diffusion of the diamagnetic molecules and ceasing the diffusion of paramagnetic molecules. In Section 2.3, we analyze the role of the magnetic concentration-gradient force in the diffusion of hemoglobin and oxygen in red blood cells (RBCs) as well as the diffusion of some biologically active molecules. The achieved results and prospects for further research are discussed in Section 3.

## 2. Results

### 2.1. Magnetic Concentration-Gradient Force

Let us consider a case when a uniform static magnetic field (∇*B* = 0) is applied to a cell. In this case, the magnetic concentration-gradient force acts on diamagnetic and paramagnetic molecules and can either assist or oppose ion movement through the cell. The volume density of the concentration-gradient magnetic force is given by [23]:(1)$$\vec{f}=\frac{\chi B^{2}}{2\mu_{0}}\vec{\nabla }n$$
where *n* is the molar concentration of molecules with the molar magnetic susceptibility *χ*, and *μ*_0_ is the vacuum permeability. In a particular case when the concentration only depends on one coordinate, *x*, the force is determined as follows:(2)$$f\left(x\right)=\frac{\chi B^{2}}{2\mu_{0}}\frac{dn\left(x\right)}{dx}$$

To analyze the influence of this magnetic force on the diffusion of paramagnetic or diamagnetic molecules, we calculate the force exerted per molecule, *f*_1_ = *f/(nN_A_)*. From Equation (1), this force reads as:(3)$${\vec{f}}_{1}=\frac{\chi B^{2}}{2\mu_{0}nN_{A}}\vec{\nabla }n$$

Under this force (Equation (3)), the flow velocity of a paramagnetic or diamagnetic solute is *u* = γ*f*_1_, where γ is the mobility of a diffusing molecule in a solution. Depending on the sign of the magnetic susceptibility (*χ* < 0 for diamagnetic and *χ* > 0 for paramagnetic species), the force is either parallel or antiparallel to the concentration gradient.

Let us compare the concentration-gradient magnetic force density (Equation (1)) with the magnetic gradient force density $f_{grad}=\frac{\chi n\;B}{\mu_{0}}\nabla B$ [23]; for example, in MRI scanners magnetic gradient values are typically between 5 and 50 mT m^−1^ [32], and the magnetic induction is between 3 and 7 T. The ratio between these forces is $\frac{f_{grad}}{f}=\frac{2\nabla B}{B}\frac{n}{\nabla n}$. Thus, in commercially available worldwide MRI scanners, the ratio ∇*B*/*B* » (10^−3^–10^−4^) m^−1^. In living cells, the value of the ratio *n*/∇*n* is also small, e.g., for hemoglobin it is *n*/∇*n* = *R_c_* » 4 μm = 4 × 10^−6^ m (see Section 2.3.1). Hence, in MRI setups, the ratio between the magnetic gradient force and the concentration-gradient magnetic force is in the order of 10^−9^, which implies that the magnetic gradient force is negligible compared to the concentration-gradient magnetic force. In such a case, the concentration-gradient magnetic force is operative in the human body during MRI scanning. In an inhomogeneous magnetic field, ~1 T in aqueous electrolytes, the gradient magnetic force was found to be 10–10^3^ times larger than the Lorentz force and the magnetic gradient force [23].

It is shown below that the static magnetic field can directly affect diffusion. Several effects related to the role of the force (Equation (3)) in diffusion are considered in detail below.

### 2.2. A Magnetic Field’s Effect on the Diffusion of Paramagnetic and Diamagnetic Molecules

The differential equation that describes the diffusion of molecules in a solution moving with velocity *u* is described as [23,33,34,35]:(4)$$\frac{\partial n}{\partial t}=D\nabla^{2}n-\left(\vec{\nabla }\cdot n\vec{u}\right),$$
where *D* is the diffusion coefficient. Note that Equation (4) is valid in the diffusion approximation [34].

Substantiating *u* = *γf*_1_ into Equation (3), one can arrive at:(5)$$\frac{\partial n}{\partial t}=D_{eff}\nabla^{2}n,$$
where $D_{eff}=\left(D-\frac{\gamma \chi B^{2}}{2\mu_{0\;}N_{A}}\right)$ is the effective diffusion coefficient. Given that the mobility (γ) of the diffusing molecule in a solution may be calculated using the Nernst–Einstein relation, γ = *D/k_B_T* (where *k_B_* is the Boltzmann constant and *T* is the temperature), one can arrive at:(6)$$D_{eff}\left(B\right)=D\left(1-\beta \right),$$
where
(7)$$\beta =\frac{\chi B^{2}}{2\mu_{0}RT}$$
and *R* = 8.31 J/(K mol) is the gas constant.

Thus, the diffusion process affected by the concentration-gradient magnetic force is described by Equation (5) with *D_eff_*(*B*), which is a partial differential equation. As seen from Equations (6) and (7), an MF decreases (*β* > 0) the diffusion coefficient for paramagnetic molecules (*χ* > 0) and increases it (*β* < 0) for diamagnetic molecules (*χ* < 0). The case *β* << 1 describes the effects of weak magnetic fields on diffusion. The parameter, *β* ≈ 1 corresponds to the suppression of the diffusion of paramagnetic molecules when the concentration flux (Fick’s law) and magnetic flux compensate each other.

The mechanism of the suppression of the diffusion process is qualitatively explained as follows. In a diffusion process, particle flux due to the gradient in concentration is directed from the highest concentration to the lowest concentration, *J_D_* = −*Ddn/dx* (Fick’s law), which is antiparallel to ∇*n*. Note, this flux is driven by the second law of thermodynamics.

In the presence of magnetic fields, a magnetically driven flux (*J_mag_*) arises to decrease the magnetic energy of the system (which is negative with the volume density −*c*(*pB*), where ***p*** is the vector of the particle’s magnetic moment, and *c* is the volume concentration of paramagnetic solute) and tends to move paramagnetic molecules towards an area with highest particle concentration. Indeed, the magnetic concentration-gradient force is parallel to ∇*n* as noted in Equation (1). Thus, the magnetic flux (*j_mag_*) is driven by energy minimization and is parallel to ∇*n* (Figure 1a). Since the *J_D_* and *J_mag_* fluxes are antiparallel, the diffusion rate is decreased (for an illustration, see Figure 1a).

In the case of diamagnetic solute (*χ* < 0), the magnetic force (Equation (1)) is antiparallel to the concentration gradient and, therefore, the *J_D_* and *J_mag_* fluxes are parallel (Figure 1b). This finding implies that a magnetic field accelerates the diffusion of diamagnetic molecules. However, since diamagnetic susceptibilities are several orders of magnitude less than paramagnetic susceptibilities, the magnetic effect on diffusion is rather small. However, since *β*∝*T*^−1^ (Equation (7)), the magnetic contribution to the diffusion increases as the temperature decreases. Thus, at low temperatures, one can expect a noticeable effect of high magnetic fields on the diffusion rate of diamagnetic species.

A 3D diffusion problem with spherical symmetry is described by the following equation:(8)$$\frac{\partial n}{\partial t}=D_{eff}\left(B\right)\left(\frac{\partial^{2}n}{\partial r^{2}}+\frac{2}{r}\frac{\partial n}{\partial r}\right),$$
where *r* is the radial coordinate.

To solve the differential Equation (8), we chose the simplest boundary and initial conditions:(9)$$n\left(t,0\right)=0,\;n\left(t,R\right)=n_{0}\;and\;n\left(0,r\right)=g\left(r\right),$$
where *n*_0_ is the constant concentration at the surface of the sphere, *r* = *R*, and the function *g(r)* is the initial concentration inside the sphere. Our choice of boundary conditions is determined by biologically relevant diffusion problems in cells and tissues. For example, the differential equation describing the ATP diffusion between a source and sinks was solved under the boundary conditions fixing the ATP concentrations at two spatially separated planes in work [31]. Boundary conditions of this type are applicable in cell biophysics to solve the following problems: diffusion through a cell membrane, lateral diffusion of proteins in membranes, intracellular signaling and calcium dynamics, drug diffusion through tissue, diffusion of transcription factors and other DNA-binding proteins along DNA, and gas diffusion in erythrocytes. Below, we apply Equation (8) with the imposed boundary conditions (Equation (9)) to the analysis of the diffusion of hemoglobin and oxygen in red blood cells.

The solutions to Equations (8) and (9) are described by the equation [36]:(10)$$n\left(t,r\right)=n_{0}\left(1+\frac{2R}{\pi R}\overset{\infty }{\underset{n=1}{\sum }}\frac{{\left(-1\right)}^{n}}{n}Sin\left(\frac{n\pi r}{R}\right)Exp\left[-\frac{D\left(1-\beta \right)\pi^{2}n^{2}}{R^{2}}t\right]\right)$$

In Figure 2a–c, we show the solution for Equation (10) as the density plots of *n(t*,*r)/n*_0_, calculated using Wolfram Mathematica 10 software [37], for the three cases: diffusion with no magnetic field (*β* = 0) and diffusion in a magnetic field (*β* = 0.5 and *β* = −0.5). The calculations of *n(t*,*r)/n*_0_ were performed for *R* = 5 μm, *D* = 10^−9^ m^2^/s which is the O_2_ diffusivity inside a red blood cell (RBC) [38].

The total amount of diffusing substance entering or leaving the sphere is given in the form [36]:(11)$$\frac{M\left(t\right)}{M_{\infty }}=1-\frac{6}{\pi^{2}}\sum_{n=1}^{\infty }\frac{1}{n^{2}}Exp\left[-\frac{D\left(1-\beta \right)\pi^{2}n^{2}}{R^{2}}t\right],$$
where *M*_∞_ is the corresponding quantity after infinite time.

In Figure 2d, we plotted the functions *M(t)/M*_∞_ for *β* = 0 (no magnetic field), *β* = 0.5 (paramagnetic species in an MF), and *β* = −0.5 (diamagnetic species in an MF). As seen in Figure 2d, the diffusion rate, *dM/dt* (which is the curve slope) crucially depends on the magnetic field: diamagnetic molecules in an MF *dM/dt* increases, while for paramagnetic molecules this quantity decreases.

Thus, the magnetic field can significantly inhibit the diffusion of paramagnetic molecules (see Figure 2b,d), e.g., O_2_; FeCl_3_; deoxyHb; intracellular and intercellular free radicals, such as O_3_, NO, and NO_2_; paramagnetic vesicles. On the contrary, an MF accelerates the diffusion of diamagnetic molecules (see Figure 2c,d). Below, we estimated the characteristic fields and values of the β parameter (Equation (7)) corresponding to the onset of the inhibition (or acceleration) of diffusion for paramagnetic and diamagnetic molecules.

We define the characteristic time scale of a diffusion process as *τ*_0_ ≈ *L*^2^/*D*, which appears in a solution of the equation for the diffusion in one dimension when *D* is constant, *n(x)*~*t*^−1/2^exp(−*x*^2^/4*Dt*). This solution describes the spreading by diffusion of a certain amount of substance deposited at time *t* = 0 in the plane *x* = 0. A characteristic time scale allows for the assessment of the key parameters influencing the diffusion process. In practice, a crude estimate of *τ*_0_ can be made knowing the diffusion coefficient and the length-scale (*L*) of a specific diffusion task. The characteristic time scale for diffusion in cells can be estimated from Equation (8) as follows. The characteristic time for diffusion due to the concentration gradient is $\tau_{0}\approx \frac{L^{2}}{D}$. In an MF, for paramagnetic species (*β* > 0), the characteristic diffusion time is increased as:(12)$$\tau_{eff}\approx \frac{L^{2}}{D_{eff}}=\frac{\tau_{0}}{1-\beta },$$
while for diamagnetic species (*β* < 0), this time is decreased:(13)$$\tau_{eff}\approx \frac{\tau_{0}}{\left(1-\beta \right)}.$$

In the limiting case *β*→1 (which corresponds to approaching the critical magnetic field $B_{cr}=\sqrt{\frac{2\mu_{0\;}RT}{\chi }}$), the effective diffusion time of paramagnetic molecules goes to infinity, which implies that the magnetic field completely suppresses the concentration diffusion. For, *B* > *B_cr_* the effective diffusion coefficient is negative, which means that paramagnetic species are squeezed by a strong enough magnetic field. Negative diffusion coefficients violate physical intuition because they indicate that species oppose their dilution. However, there are several ongoing discussions with regards to the existence of negative diffusion coefficients in [39,40,41,42,43] and references therein. These works give theoretical and experimental evidence supporting the existence of negative diffusion coefficients. In our case, it is possible to stop diffusion if the diffusion and magnetic fluxes are antiparallel and equal in magnitude, as these two fluxes will cancel each other out at the critical value of the magnetic induction, *B_cr_* (see Figure 1a for an illustration). Indeed, the diffusion and magnetic fluxes cancel each other out when the magnetic driving force (Equation (1)) is equal to the diffusion driving force, $\vec{f_{D}}=RT\vec{\nabla }n$ [23]: $\frac{\chi B^{2}}{2\mu_{0}}\vec{\nabla }n=RT\vec{\nabla }n$. It directly follows from this equation that *D_eff_* = 0 at *β* = 1 when approaching the critical magnetic field *B_cr_*.

By setting the change of the characteristic diffusion time (*τ_eff_* − *τ*_0_)/*τ*_0_ = *β*/(1 − *β*) = ±5%, in Equations (12) and (13), we defined the parameter *β*_0_ ≈ 0.05 as the value of the onset of an MF effect on diffusion.

Since diffusion processes determine the reference time scale for all other processes in cells as well as the propagation speed of signaling molecules during cell-to-cell communication [44], our results—the magnetic field dependences of the diffusion rate and characteristic time—may serve as an important key for revealing and understanding the mechanisms of magnetic field action on living cells, tissue, and organisms.

### 2.3. Diffusion of Biologically Active Molecules in Specific Examples of Biomedical Applications

In Table 1, for several biologically active molecules, we present the MF induction values corresponding to *β*_0_ = 0.05 at which an MF starts to affect diffusion as calculated from $B_{0}=\sqrt{\frac{2\mu_{0\;}\beta_{0}RT}{\chi }}$. Below, we analyze the MF’s effects on diffusion for some biologically active molecules, such as oxygen and hemoglobin as well as molecules used as contrast agents in MRI and anticancer paramagnetic drugs.

#### 2.3.1. Diffusion of Hemoglobin and Oxygen in Red Blood Cells

The diffusion of hemoglobin and oxygen inside an RBC drives the oxygen uptake and release by RBCs. Since deoxyhemoglobin (deoxyHb), methemoglobin (metHb), and oxygen (O_2_) are paramagnetic species (see Table 1), the diffusion rates of O_2_, deoxyHb, and metHb decrease in an MF as shown in Figure 2b. The diffusion coefficients of hemoglobin and oxygen at concentrations existing inside RBC are important parameters for quantitative description of oxygen uptake and release by RBCs. The mean diffusion time of Hb inside an RBC is on the order of *τ_Hb_* = *a*^2^/*D_Hb_* ≈ 1 s, where *D_Hb_* ≈ 1.6 × 10^−11^ m^2^/s [49] or *D_Hb_* ≈ 3.4 × 10^−12^ m^2^/s [50]. A large difference in the values of *D* could be explained as follows. Cell is a living system, which is able to adapt its physical parameters to the alternating environment, for example, by adjusting the diffusion coefficient which, in turn, can be modulated through conformational distributions in the protein. For instance, for LacI repressor proteins diffusing along with DNA, the measured 1D diffusion coefficients were found to vary in a large range, from 2.3 × 10^−12^ to 1.3 × 10^−9^ cm^2^/s [51]. The oxygen diffusion time inside an RBC is on the order *τ_ox_* = *a*^2^/*D_ox_* ≈ 10 ms, where *a* ≈ 3 μm is the RBC radius, and *D_ox_* ≈ 10^−9^ m^2^/s is the O_2_ diffusivity inside an RBC [38]. Thus, the characteristic time of oxygen diffusion is short compared with the time Δ*t_trap_* associated with O_2_ trapping by Hb, which is a few tens of milliseconds [52,53]. Thus, the dynamics of oxygen capture by RBCs is characterized by the following time hierarchy: *τ_ox_* (≈10 ms) < Δ*t_trap_* (≈10–100 ms) < *τ_Hb_*(≈1 s). Of note here, Hb diffusion is the slowest process, and its characteristic time is comparable with the time that erythrocytes spend in the lung alveoli ≈800 ms [54,55,56]. Bearing in mind that the Hb diffusion time could be increased by a sufficiently high MF (see Table 1), one can conclude that a 20 T magnetic field application may decrease the blood saturation by oxygen if *τ_Hb_* becomes greater than 800 ms. As mentioned above, ~20 T MFs are currently used in ultra-high-resolution MRI scanners.

Diffusion is closely related to cell homeostasis. Indeed, when homeostasis is threatened in a cell, diffusion helps to maintain balance in cellular concentrations. Thus, an MF applied to a whole cell body can disturb its dynamic equilibrium (homeostasis) and change both the cell’s shape and its volume. The concentration-gradient magnetic force (Equation (1)) pushes Hb molecules towards the RBC membrane and creates a magnetic pressure on it. In fact, the RBC membrane is impermeable to hemoglobin. Knowing the volume density of the magnetic force (Equation (2)), one can calculate the magnetic pressure of Hb on the cell membrane as *P* = *fV*/*S*, where *V* is the cell volume and *S* is the membrane area. For a spherical cell, one can arrive at the following equation:(14)$$P_{mag}=\frac{\chi R_{c}B^{2}\nabla n}{6\mu_{0}},$$
where *R_c_* is the mean cell radius. For deoxyHb, the magnetic susceptibility is *χ_Hb_* = 60.4 × 10^−8^ (m^3^/mol) [45]. Magnetic susceptibilities for Hb and metHb are *χ_Hb_* = −4.754 × 10^−7^ (m^3^/mol) and *χ_metHb_* = 7.217 × 10^−7^ (m^3^/mol) [46], respectively. The other relevant parameters of Hb include the molecular weight of deoxyHb is M_Hb_ = 64,450 g/mol [57], the molar volume of Hb in solution is v_m_ = 48,277 mL/mol [58], and the Hb concentration is n_Hb_ = 5.5 mol/m^3^ [59]. Equation (12) describes the ability of the magnetic pressure to disturb the cell’s pressure balance and change the RBC volume. The RBC volume magnetic susceptibilities were determined by SQUID [60]: *χ_RBC_* = −9.23 × 10^−6^ (oxy RBC), −5.72 × 10^−6^ (deoxy RBC), and −5.27 × 10^−6^ (met RBC) in the SI units system.

Now, we estimate the magnetic pressure exerted on deoxyHb in RBCs. The above shown Hb susceptibility and intracellular concertation, *R_c_* = 4 μm, and the volume averaged concentration gradient, ∇*n* = *n_nb_/R_c_* (a layer of absorbed deoxyHb is located on the inner surface of an erythrocyte as depicted in Figure 3), for a completely deoxygenated RBC in a static MF with B = 100 T are inserted into Equation (14). Thus, we obtain the magnetic pressure, *P_mag_* ≈ 4000 Pa. We suppose that a magnetic pressure of hundreds Pa magnitude can disturb the cell pressure balance determined by osmotic pressure, hydrostatic pressure, and membrane and cortical tension [61]. For example, mitotic HeLa cells exert a rounding pressure (pressure generated by mitotic cells to create space to divide) of 10–150 Pa [62].

As noted in Equation (14), the magnetic pressure is proportional to the square of magnetic induction and the concentration gradient of the deoxyHb. We conclude that in a sufficiently high magnetic field, a completely deoxygenated red blood cell exhibits swelling due to the magnetic pressure (Figure 3), while a completely oxygenated red blood cell does not change its volume. Thus, the swelling effect could be well pronounced only for completely deoxygenated erythrocytes. Despite this findings, the cell membrane integrity is preserved with a strain as high as 40–50% [63]. In sufficiently high magnetic fields (~100 T), magnetic pressure (Equation (14)) and swelling can cause RBC membrane rupture.

It should be noted here that regardless of the above described swelling mechanism, magnetically induced swelling and shape changing of cells (with no paramagnetic agents) have been observed under the following conditions: (i) osteoblasts adhered to glass substrates in the presence of a 5 T magnetic field [64] and (ii) human THP-1 leukemia cells (volume increases up to 90%) after 24 h of exposure to a spatially modulated high-gradient magnetic field [65].

An interesting aspect of magnetically induced RBC swelling that deserves further study is the diffusion of oxygen (O_2_) through the RBC membrane. Given that O_2_ is paramagnetic, the concentration-gradient magnetic force (Equation (1)) pushes O_2_ molecules either outside or inside an RBC depending on the direction of the gradient ∇n_O2_, thereby facilitating/complicating oxygen transport in capillaries. Moreover, mitochondrial function in cells is also crucially dependent on the diffusion rate of O_2_ molecules, which could be decreased by a sufficiently high magnetic field (see Table 1).

#### 2.3.2. Diffusion of Molecules Used in Medicine and Paramagnetic Drugs

The study of an MF’s effects on the diffusion of paramagnetic molecules used as contrast agents in MRIs and paramagnetic drugs is especially important in the field of diagnostics and therapeutics. For example, FeCl_3_-induced arterial thrombosis is used to control bleeding from damaged blood vessels. The experiments [66] showed that ferric chloride (FeCl_3_) compared to the current standard method, which includes suturing the bleeding site, requires significantly less time to control bleeding with even the lowest concentration of ferric chloride. Ultrastructural analysis revealed that FeCl_3_ diffused through the vessel wall, resulting in endothelial cell denudation without exposure of the inner layers [67].

Let us analyze how diffusion of FeCl_3_ through the blood vessel wall can be affected by an MF. Ferric chloride is paramagnetic with a magnetic susceptibility *χ* = +1.345 × 10^−8^ m^3^/mol (see Table 1). As described in Section 2.2, a sufficiently high MF will hinder and slow FeCl_3_ diffusion. Thus, magnetically hindered thrombosis with FeCl_3_ will require a greater concentration of FeCl_3_ compared with its application with no magnetic field. The effects of an MF on FeCl_3_-induced thrombosis were studied in rat model experiments [68]. In the group of rats exposed to a static 600 mT magnetic field thrombus protein content (0.28 ± 0.14 mg/mL) was downregulated compared with the model control group (0.47 ± 0.10 mg/mL). This MF effect was putatively explained by MF-induced vascular smooth muscle relaxation and blood viscosity reduction. In light of that mentioned above, it is unlikely that the observed decrease in thrombus protein content in 600 mT, MFs could be caused by impaired diffusion of FeCl_3_ to the bleeding site. Indeed, as noted in Table 1, MFs with induction greater than one hundred teslas affect FeCl_3_ diffusion.

The paramagnetic ion gadolinium Gd(III) is commonly used in MRIs as contrast agents to provide increased contrast [2,69]. Importantly, this agent is paramagnetic with a sufficiently high magnetic susceptibility, *χ* = 18.5 × 10^−8^ m^3^/mol, see Table 1. Thus, for MFs with induction *B* < 40 T, the MF effect on Gd diffusion is expected to be negligible. Thus, no inexpedient diffusional effects on the required distribution of Gd contrast agent are observed from an MF in an MRI machine.

Two paramagnetic species with the highest magnetic susceptibility values are found in the last two rows of Table 1: MnCl_2_ and Ho(NO_3_)_3_. Manganese chloride is used as a nutraceutical and an MRI contrast agent. Holmium (Ho^166^) and holmium nitrate are used in medicine for various therapeutic applications including intratumoural cancer treatment, targeted therapies, skin patches, bone-seeking agents in bone marrow transplantation, and selective internal radiation therapy (SIRT) [70]. After injection of a drug into a specific site or organ, the drug concentration decreases due to the fact of diffusion and biodistribution. Thus, the drug efficacy decreases with time. Therefore, the following important question arises: Is it possible to prevent drug diffusion from the target with an MF. The answer is probably yes. However, this requires applications of MFs with an induction of 50–100 T (see Table 1). Importantly, such a high MF would prevent common drug side effects related to undesirable drug biodistribution outside the target site. Of note, as mentioned in the introduction, currently, static 40–50 T magnetic fields are reachable in the lab, while MFs of a thousand teslas are only attainable using a pulsed regime.

## 3. Conclusions

We analyzed the roles of static MFs and the magnetic concentration-gradient forces in diffusion of paramagnetic and diamagnetic molecules. It was shown that a high magnetic field accelerated the diffusion of diamagnetic molecules and reduced the diffusion of paramagnetic molecules. We revealed the underlying mechanisms of magnetically affected diffusion and the key parameter that determines the strength of an MF’s effect on diffusion, namely, the ratio between the magnetic field’s energy and that of thermal fluctuations (see Equation (7)). The proposed mechanisms and the predicted MF’s effect on diffusion require further theoretical and experimental investigation. For example, in living cells, magnetically induced changes in the diffusion coefficient could be detected experimentally using an approach based on electron spin resonance microscopy [71].

We theoretically predicted a new biological effect of homogeneous high magnetic fields: magnetically induced swelling of deoxygenated red blood cells. A deoxygenated erythrocyte contains deoxygenated hemoglobin, which is paramagnetic. In a high magnetic field, due to the large radial gradient of deoxyHb, the magnetic pressure (Equation (12)) exerted on deoxyHb causes erythrocyte swelling as depicted in Figure 3. As the erythrocyte volume increases, the surface of its membrane also increases. The increase in the membrane area subsequently leads to an increase in the amount of oxygen molecules diffusing into the erythrocyte. Through a chain of chemical reactions, these oxygen molecules bind to Hb making it diamagnetic. Then the magnetic pressure falls to zero, and the initial volume is restored in the erythrocyte. RBC swelling and shape alternations in MFs could affect (probably hinder) oxygen transport from capillaries to tissue. Experimental verification of the prediction of magnetically induced RBC swelling could answer the following important question: how does a high magnetic field (e.g., in ultra-high-resolution MRI scanners) affect human health?

Our study gleaned new insights into the potential biomedical applications of high and ultra-high magnetic fields and provided new perspectives on the control of diffusion in cells and tissues. In terms of future directions for the development of magnetotherapy [72], our findings argue that cell-to-cell communication can also be directly affected by MFs. Indeed, the propagation time of chemical signals across the cell envelope, extracellular matrix, and tissue is limited by diffusion. Therefore, changes in characteristic diffusion time and subsequent cell signaling alterations induced by an MF represent an intriguing perspective to improve the understanding of molecular signaling pathways and development of new treatment approaches for cell therapy. Thus, the application of the elaborated diffusion model and MFs is of vital importance for future studies of cell signaling processes, cell therapy, disease modeling, tissue regeneration, magnetic surgery [73], and tissue engineering using magnetic fields [74].

## Figures and Tables

**Figure 1 cells-11-00081-f001:**
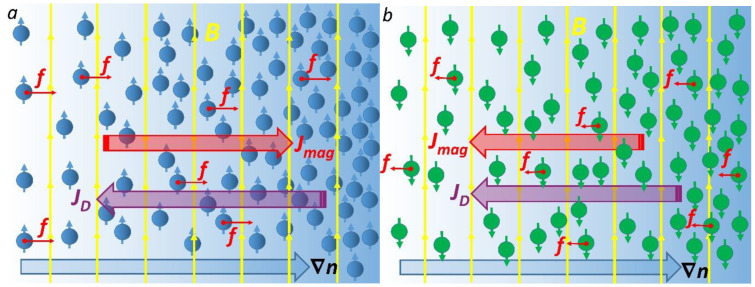
Sketch of the magnetically affected diffusion of paramagnetic (**a**) and diamagnetic molecules (**b**). The yellow arrows represent the magnetic field lines. The large blue arrow shows the concentration gradient ∇*n* of solute. Red arrows show the magnetic concentration-gradient forces (Equation (1)) acting on the molecules.

**Figure 2 cells-11-00081-f002:**
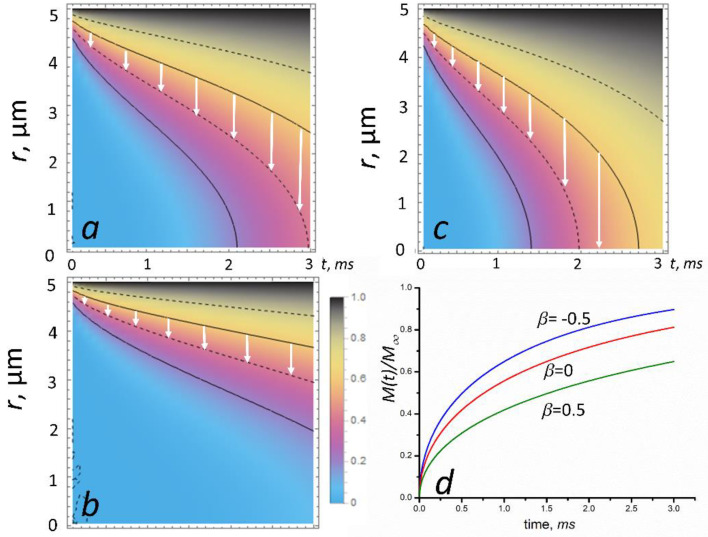
A MF’s effect on diffusion in a sphere: the contour plots of *n(t*,*r)/n*_0_ given by Equation (10). The sphere is initially at a uniform zero concentration and the surface concentration is maintained constant at *n*_0_. The curves represent *n(t*,*r)/n*_0_: (**a**) with no magnetic field (*β* = 0); (**b**) with a magnetic field corresponding to *β* = 0.5 (paramagnetic species); (**c**) with a magnetic field corresponding to *β* = −0.5 (diamagnetic species). The calculations of *n(t*,*r)/n*_0_ were performed for *R* = 5 μm and *D* = 10^−9^ m^2^/s in the time interval 0 < *t* < 3 ms. The legend shows the concentration *n(t,r)/n*_0_, which varies from 0 to 1. The white arrows show the direction of the diffusion front propagation. Figure (**d**) shows the total amount of diffusing substance entering or leaving the sphere as a function of time for *β* = 0, −0.5, and 0.5.

**Figure 3 cells-11-00081-f003:**
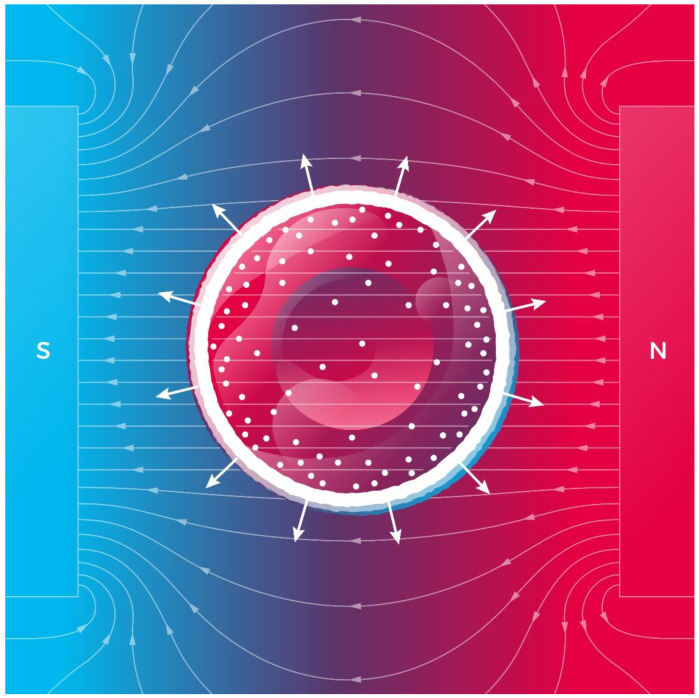
Scheme of a deoxygenated red blood cells under magnetic pressure on deoxyHb. The white radial arrows show directions of the magnetic concentration-gradient forces. Small white circles represent deoxyHb molecules, while the large white circle represents a layer of absorbed deoxyHb on the inner surface of the membrane of an RBC.

**Table 1 cells-11-00081-t001:** Values of the magnetic field induction (in teslas), corresponding to the onsets of the MF’s effects on diffusion, as calculated from Equation (7) for different paramagnetic and diamagnetic molecules and *β*_0_ = 0.05. Gray colors indicate diamagnetic molecules.

Molecules	*χ*, m^3^/mol, References	*B*_0_, T
deoxyHb	+60.4 × 10^−8^ [45]	22.8
metHb	+7.217 × 10^−7^ [45,46]	20.8
oxyHb	−4.754 × 10^−7^ [45,46]	25.7
O_2_	+4.3 × 10^−8^ [47]	85.3
Gd	+18.5 × 10^−8^ [47]	41.2
FeCl_3_	+2.573 × 10^−8^ [48]	110
MnCl_2_	+3.8 × 10^−8^ [48]	90.8
Ho(NO_3_)_3_	+11.34 × 10^−8^ [48]	52.6

## Data Availability

Not applicable.

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
