# Peer review of "Effects of High Magnetic Fields on the Diffusion of Biologically Active Molecules"

_cells, 2021, doi:10.3390/cells11010081_

Round 1

Reviewer 1 Report

The article is concise and well written on an important topic. The results show that concentration-gradient magnetic force can accelerate diffusion of diamagnetic molecules and stop the diffusion of paramagnetic molecules.

Minor suggestions:

- For the article, a section/paragraph may be useful, which discusses what minimum magnetic field gradients can be realized in real experiments with generators of a high magnetic field in order to consider only the discussed force component and related effects?

- What are the relative contributions of the two components of the magnetic force associated with the spatial field gradient and concentration gradient in real experiments or in typical MRI setup?

- The abbreviation ROS is used several times in lines 13, 84, 192, but the full notation as "reactive oxygen species (ROS)" is given only in line 282.

- There is no full bibliographic data in the reference #23 (line 562). Accessed date (Date Month Year) should be given to the web reference # 37 (line 588)

Author Response

Authors’ Responses

We are very grateful to the reviewer for positive and constructive comments and suggestions to our manuscript.

The reviewer’s comments are given in black, while authors’ responses are given in blue.

Reviewer 1

The article is concise and well written on an important topic. The results show that concentration-gradient magnetic force can accelerate diffusion of diamagnetic molecules and stop the diffusion of paramagnetic molecules.

Minor suggestions:

  1. - For the article, a section/paragraph may be useful, which discusses what minimum magnetic field gradients can be realized in real experiments with generators of a high magnetic field in order to consider only the discussed force component and related effects?

Thanks for this question. There is no defined minimum field gradient value. In each specific magnetic system that generates a strong static magnetic field, the field appears as an inhomogeneous (gradient) magnetic field. However, inside a sufficiently long solenoid (in its middle part), it is always possible to select a region with a quasi-uniform field with the smallest value of the magnetic gradient (in an ideal case, zero gradient area). For example, in experiments [1] with mice in a superconducting magnet that generates a static magnetic field 33 T, the gradient was zero in the middle part of the magnet and 40-100 T/m between the magnet center and edge.

  1. - What are the relative contributions of the two components of the magnetic force associated with the spatial field gradient and concentration gradient in real experiments or in typical MRI setup?

Thank you for this issue.  First, we would like to point out that in gradient magnetic fields, biomagnetic effects can be expected in cells/organisms exposed to static magnetic fields with gradients exceeding 103-104 T m-1 [2]. In MRI machines, a magnetic field gradient sets different precession frequencies across the imaging volume, which makes it possible to control the spatial distribution of magnetization. In MRI setups gradient values are typically between 5 and 50 mT m−1[3], which are 6-7 orders of magnitude smaller than the abovementioned gradient.

Let us compare the concentration-gradient magnetic force density     (Eq. 1)  with the magnetic gradient force density     [4].   The ratio between them is     .  In MRI machines the ratio ÑB/B» (10-3—10-4) m-1 , while in living cells the ratio n/Ñn is, e.g. for the hemoglobin  n/Ñn=Rc »4 mm = 4 10 -6m  (see the MS text).  Thus, in a typical MRI setup, the ratio between the magnetic gradient force and the concentration-gradient magnetic force is in the order of 10-9, which implies that the magnetic gradient force is negligible compared to the concentration-gradient magnetic force.

In the revised MS, in Section 2.1, we have added a discussion on this point.

3.- The abbreviation ROS is used several times in lines 13, 84, 192, but the full notation as "reactive oxygen species (ROS)" is given only in line 282.

Thanks. Corrected.

4.- There is no full bibliographic data in the reference #23 (line 562). Accessed date (Date Month Year) should be given to the web reference # 37 (line 588)

Thanks. Corrected.

  1. Tian, Xiaofei, Yue Lv, Yixiang Fan, Ze Wang, Biao Yu, Chao Song, Qingyou Lu, Chuanying Xi, Li Pi, and Xin Zhang. "Safety Evaluation of Mice Exposed to 7.0–33.0 T High-Static Magnetic Fields." Journal of Magnetic Resonance Imaging 53, no. 6 (2021): 1872-84.
  2. Zablotskii, V., O. Lunov, S. Kubinova, T. Polyakova, E. Sykova, and A. Dejneka. "Effects of High-Gradient Magnetic Fields on Living Cell Machinery." Journal of Physics D: Applied Physics 49, no. 49 (2016): 493003.
  3. McRobbie, D. W. "3.01 - Fundamentals of Mr Imaging." In Comprehensive Biomedical Physics, edited by Anders Brahme, 1-19. Oxford: Elsevier, 2014.
  4. Hinds, G., J. M. D. Coey, and M. E. G. Lyons. "Influence of Magnetic Forces on Electrochemical Mass Transport." Electrochemistry Communications 3, no. 5 (2001): 215-18.

Reviewer 2 Report

The manuscript is devoted to the study of the possibility of controlling diffusion processes in a biological cell using a strong magnetic field (MF), at a level from tens to thousands of Tesla. The manuscript is of interest. There are following comments.

The authors consider a wide range of MFs from tens to thousands of T, but do not compare the possibility of widespread use of such fields with their cost. The generation of a constant MF with an intensity of only 15 T in a volume of several liters is expensive. In the 1960s, to power the solenoid, it required DC generators placed in a three-story building sized ​​35 m by 13 m. Now such devices have become more compact due to the development of superconductivity technology, but they remain expensive and complex installations. Generation of pulsed MFs is cheaper, but they cannot be used to realize the effect of MFs on diffusion in biological cells, since such strong MF pulses induce strong eddy currents that can lead to the breakdown of biological membranes and cell death.

If to consider a static MF, is it possible to separate the biological effects, possibly caused by the action of MF on the diffusion coefficient, from biological effects that inevitably arise in fields already at a level of several T, according to magnetochemical mechanisms? The authors do not discuss this important circumstance, which makes experimental observation of the claimed effect hardly possible. Then, the proposed theory does not satisfy the methodological principle of verifiability.

Writing down the diffusion equation and its components, the authors define the concentration n as “concentration of diamagnetic and paramagnetic species” (line 64), as “diamagnetic and paramagnetic ions” (line 90), as “molar concentration of ions” (line 96), and as concentration of molecules (line 103 and 116). It would make sense to give a single clear definition of the object under study. If, for instance, these particles were neutral diamagnetic particles in water, then how would the authors take into account the diamagnetic susceptibility of water? If these were ions, then it would be necessary to compare the declared “concentration-gradient magnetic effect” with the action of the Lorentz force, which markedly influences diffusion already at a MF of 1 T.

The authors used the differential equation of diffusion to describe the phenomena in a small volume of a biological cell without analyzing the limitations of such a description. In particular, the notion of a physically small element dx should make sense. In other words, dn/dx must exist, i.e. an element of volume dx*dy*dz must contain a sufficient number of particles dn >> 1. In addition, the applicability of the diffusion approximation is limited by the requirement that the deviation from thermodynamic equilibrium be small, i.e. |nabla(n)| << n/lambda, where lambda is the particle's mean free path.

To estimate the above two limitations, realistic values of the concentration of the considered particles in a biological cell or in the volume of assumed diffusion are required. The manuscript does not seem to contain such data.

In conclusion, this manuscript looks incomplete and requires additional research, revision, and subsequent peer review.

Author Response

Authors’ Responses

We are very grateful to the reviewer for positive and constructive comments and suggestions to our manuscript.

The reviewer’s comments are given in black, while authors’ responses are given in blue.

Reviewer 3 Report

This manuscript analytically establishes and solves the equations for the purpose of elucidating diffusion activities of molecules under high-intensity magnetic field exposure. The outcome reveals that a high MF accelerates diffusion of diamagnetic species while slowing the diffusion of paramagnetic molecules in cell cytoplasm. The scope of the manuscript fits for the SI. Major concerns: 1)Since I have very few experience for reviewing the submissions to Cells, I am not sure if an independent section for discussion is needed. In any case, I recommended that authors could make further insightful remarks if applicable. In the current version, many details in the analytic research can be elaborated and clarified, e.g., impact of the simplification (boundary condition, configuration of the specific parameters) to the results, and to the resultant physiological effect. Validation of the calculation results, by published articles, can also be performed in that section. 2) The difference between the previous articles should be addressed. As the canonical biophysical equations have been applied, has any novel configurations involved ? 3) Quantitative analysis is suggested to be enhanced as the authors mentioned effects by MF of various levels in introduction. Minor points: -In physics, when we talked about tesla, it is the unit for magnetic flux density. Please use the same expression throughout the manuscript. There are diverse expressions for it. -”Static magnetic fields with a magnetic in-duction 17.6-21 T are used in MRI instruments for better image resolution and more accurate diagnosis [1, 2].” However, MRI with such high intensity has not been applied in clinics by now. The application diagnosis is not authorized. -When an abbreviation is defined, it is unnecessary to redefine it. See line 59 of page 2, and line 89 of the same page

Author Response

(The authors gave the same response as above.)

Round 2

Reviewer 2 Report

The authors answered the questions of the referee. However, some issues were commented not clearly enough, in a polemic manner, and unfortunately have not been reflected in the revised manuscript.

     It is worth drawing the attention of the authors to the fact that I am not a unique reader. While reading the article, many will ask the same questions. At the same time, the authors will no longer have the opportunity to enter into polemics with the reader. Therefore, it makes sense to work out the comments to the maximum and give the information that seems necessary even to one reviewer.

     The article under review is devoted to a certain effect assumed in a constant MF. For this reason, the mention of pulsed fields is inappropriate, distracting. In addition, estimates of the critical MF in Table 1 exceeding 100 T look defiant because of their unreality as a constant MF in laboratory conditions. It is better to remove them and leave a few that could actually be used now or in the near future.

     Magnetochemistry includes spin chemistry, which examines the effect of MF on the rate of chemical reactions through the effect on the spin. There are many works and a detailed review by Steiner and Ulrich in Chem. Rev. 89:51, 1989. These effects become significant in the MF of the order of hundreds of millitesla and above. Therefore, if the authors talk about biology and MF effects in the cell, then it is not clear how in the experiment to separate the intended effect and this magnetochemical spin effect. It makes sense to comment on this circumstance in the article and change the text where necessary. In addition, when proposing a new theoretical result intended for practice, authors usually propose a specific way to test it experimentally, not being limited to a statement that it is simple.

     The authors, in fact, did not answer the question of taking into account the diamagnetic susceptibility of water in the supposed effect of the MF influence on the gradient of diamagnetic molecules in water. From their answer, it is clear that they assume that the MF can distinguish between diamagnetic water molecules and diamagnetic molecules of a solute. However, this is unreal. For example, can it be assumed that the airflow will tilt the sparse stems of rye in a wheat field without affecting the stems of wheat? Hence, when calculating the density gradient of diamagnetic molecules in water, it is necessary to take into account all molecules. Then it will turn out to be very small when it comes to concentrations of the order of mM standard for cell chemistry. Of course, authors are free to choose what to write about, but I would advise focusing on paramagnetic molecules, the description of which also having difficulties.

     Concerning the question of the applicability of the diffusion approximation. The authors seem to have moved away from a straightforward answer. However, this issue deserves serious attention, at least a comment within the article. We are not talking about the applicability of the diffusion approximation to the cell content in general, but only to the case of low concentrations n simultaneously with a small cell volume. For example, the pH in a cell is about 7.5, i.e. the concentration of hydronium ions is about 3E-8 M. Further, 1 mole = 6E23 and 1 L = 1E15 cubic micrometer = 1E15 fL; 1 femtoL is a cubic micrometer. Then 3E-8 mole/L = 18 1/fL. Is it enough? Almost yes. A cell can be taken cube a size 10 micrometers, or a volume 1E3 fL. In order the diffusion equation be applicable to a cell, one should take dx = 1 micrometer at least or smaller, or dx*dy*dz equal to 1 fL. We can see dn/dx exists for dn=18>>1. But it is close to the limit. Now, what one could arrive to taking a paramagnetic superoxide anion as a solute? Its concentration in cell is about 3E-12 M (Ch.A. McQueen - Comprehensive toxicology, 2010, p.4472) – this is 4 orders lower than that of H+. This means only about one paramagnetic superoxide molecule per cell volume. – Does the diffusion approximation work? Even if the concentration were 1E2 times higher, the problem would persist.

     The concentrations of paramagnetic molecules in the cell are low, and this is a problem for this article. At least, this issue needs a thorough discussion in the article. The text of the article should be changed in accordance with this discussion. The article would undoubtedly benefit from being cut by a third or a quarter.
